# The social cost of investor distraction: Evidence from institutional cross-blockholding

**Vivek Astvansh**[ID][1,2,3,4]*, **Tao Chen**[5], **Jimmy Chengyuan Qu**[5]

**1** Marketing Area, Desautels Faculty of Management, McGill University, Montréal, Québec, Canada,
**2** Department of Informatics, Luddy School of Informatics, Computing, and Engineering, Indiana University, Bloomington, Indiana, United States of America, **3** Environmental Resilience Institute, Indiana University, Bloomington, Indiana, United States of America, **4** Dewey Data Inc, **5** Division of Banking & Finance, Nanyang Business School, Nanyang Technological University Singapore, Singapore, Singapore

* Vivek.astvansh@mcgill.ca

## Abstract

Institutional investors routinely hold blocks of stocks in multiple firms within an industry. While such cross-blockholding boosts a portfolio firm's financial performance, could it distract investors from attending to firm activities in a nonfinancial domain, hurting its performance in that domain? The authors answer this question in the context of corporate social responsibility (CSR). They first document that cross-held firms perform worse on social responsibility than non-cross-held firms do. A quasi-natural experiment based on mergers between institutional blockholders helps establish causality. Next and as their primary contribution, the authors demonstrate investor distraction as the mechanism. Using two proxies of distraction—EDGAR search volume and shareholder proposals on socially responsible investment—they show that the negative impact of institutional cross-blockholding on CSR mainly comes from investor distraction when investors hold multiple blocks simultaneously. By highlighting the social cost of institutional cross-blockholding, this article finds a distraction effect of institutional cross-ownership, which extends our understanding of this unique ownership structure.

## Introduction

Institutional cross-blockholding—a special ownership structure in which institutional investors simultaneously hold blocks in multiple firms in the same industry—has been extensively researched in recent years [1–4]. This research has shown that institutional cross-blockholding affects a portfolio firm's financial decisions, which in turn impacts managers' and shareholders' interests [1, 4, 5].

In comparison, academics and practitioners know little about whether, and if yes, how (that is, positively or negatively) institutional cross-blockholding affects the interests of *nonfinancial* stakeholders [6, 7]. In addition, they lack empirical evidence on why such an effect may exist— that is, the underlying mechanism [4]. This article attempts to provide this knowledge by first showing that institutional cross-blockholding stifles a portfolio firm's performance on corporate social responsibility (CSR)—hereafter, corporate social performance (CSP). However, the primary contribution of our research is in empirically demonstrating the *distraction*

**Data Availability Statement:** We merged data from the following nine sources. These databases are provided by Wharton Research Data Services. One can email wrds-support@wharton.upenn.edu and purchase subscriptions. Standard & Poor's

Compustat-Capital IQ North America Fundamentals Annual MSCI ESG KLD STATS Thomson Reuters Institutional (13F) Holding database FactSet's Institutional Holdings database Institutional Brokers Estimate System (I\B\E\S) database EDGAR Server Log File Standard & Poor's Compustat – Capital IQ Execucomp database Institutional Shareholder Services' (ISS') Shareholder Proposal database Sustainalytics database.

**Funding:** The author(s) received no specific funding for this work.

**Competing interests:** The authors have declared that no competing interests exist.

mechanism—that is, cross-blockholding distracts investors from attending to a portfolio firm's CSR. The distraction in turn stifles the firm's performance in the social domain.

Our theoretical premise is as follows. Prior research has documented that institutional investors prefer to hold stocks of socially responsible firms [8–11] and engage in the portfolio firms' CSR activities [12–14]. Invoking this logic, we expect institutional cross-blockholding to influence portfolio firms' CSP in two opposite ways.

On the one hand, institutional cross-blockholding is expected to increase firms' CSP for several reasons. First, competing firms with shared blockholders have the motivation to do more CSR activities in order to keep the institutional cross-blockholders because an exit of shared investors is a bad signal for firms [2]. Second, investors can get more information advantages and governance experience when holding simultaneously stocks of firms in the same industry [4]. Because institutional investors attach importance to firms' CSP [12], institutional cross-blockholders should be more efficient in engaging with firms' CSR activities. In addition, if institutional cross-blockholding increases portfolio firms' market share and alleviates their outside pressure on earnings [1, 15], firms may have more resources to engage in CSR activities, thus increasing their CSP.

On the other hand, because investors' attention is a limited resource [16, 17], it is less feasible for investors to keep the same effort that they put into their portfolio firms when the portfolio size becomes larger [18]. In this case, institutional cross-blockholding might cause a drop in firms' CSP because distracted investors engage less in firms' CSR activities [12]. Moreover, holding multiple firms may also change investors' allocation of attention across corporate decisions. Because defining corporate social activities and measuring CSP are not straightforward, the benefits of CSR are less tangible to investors in the short term [19, 20]. As such, under competition among other peer investors [21], institutional investors may allocate more attention to corporate decisions that directly affect their portfolio performance and focus less on firms' CSP [22]. Based on these arguments, if holding multiple blocks distracts investors' attention to CSR activities, portfolio firms are expected to perform worse in CSR under institutional cross-blockholding.

Using a comprehensive sample of U.S. listed firms during the period of 1995–2014, we examine the impact of cross-blockholding on firms' CSP (measured using KLD database). Our multivariate ordinary least squares (OLS) regression reports a negative relation between cross-blockholding and firms' CSP, which supports the distraction hypothesis. In terms of economic magnitude, firms being cross held, on average, show a .16 decrease in the overall CSP score in the following year. Next, we conduct a variety of robustness tests across different sets of fixed effects, measures of cross-blockholding, measures of CSP, and calculation methods of CSP scores. Using the sample excluding the 2008 financial crisis, we alleviate the concern that our results may be driven by firms' response to the financial crisis [23]. According to prior literature, ESG ratings from different rating agencies show low correlation, which makes the data validity questionable [23–25]. Therefore, we replicate our finding using a sample of firms available in the Sustainalytics database (in place of KLD database).

To alleviate endogeneity problems, we follow [15], and use a quasi-natural experiment based on exogenous shocks from mergers between institutional blockholders [26]. Because mergers between institutional blockholders are less likely to be driven by their portfolio firms and can only affect portfolio firms through their blockholders, this experiment provides an ideal framework to establish a causal link between institutional cross-blockholding and firms' CSP. A difference-in-differences (DID) analysis reports a significantly lower CSP in post-merger years for firms affected by investor mergers than their unaffected peers. This evidence establishes a causal link between institutional cross-blockholding and CSR. Our DID results stays when we adopt the propensity score matching to construct matched samples before the

DID analysis (PSM-DID) on alternate sets of matching variables. By decomposing CSP into CSR strengths and concerns, we observe that institutional cross-blockholding affects firms' CSP mainly by increasing CSR concerns. This finding further supports the distraction channel [12]. Next, we test the effect on five dimensions of CSP. Results suggest that institutional cross-blockholding reduces significantly firms' performance in workforce diversity, employee relations, and product quality dimensions, but not in community and environment dimensions. These results are consistent with the intuition that investors are likely to be more punitive if firms perform worse in community and environment dimensions [26]. Consequently, blockholding does not impact CSP on these two dimensions but hurts CSP on the other three dimensions.

To examine the distraction channel through which institutional cross-blockholding affects firms' CSP, we follow prior literature and conduct the following tests. First, we use EDGAR search volume as a direct measure of investor attention [27–29]. In this test, we first find that firms under greater institutional cross-blockholding receive less attention from investors after blockholder mergers and that firms with less attention before blockholder mergers decrease more in CSP under greater institutional cross-blockholding. Second, by investigating shareholder proposals on socially responsible investment (SRI), we find a significant reduction in the number and percentage of SRI proposals for firms under institutional cross-blockholding. Because shareholder proposals reflect shareholders' intention [12, 18, 30], this evidence suggests investors' decreased attention to portfolio firms' CSR decisions. Again, this finding supports the distraction hypothesis.

This article contributes to the literature in several ways. First, it adds to the literature on institutional cross-blockholding by showing its inadvertent impairment on firms' CSP. Based on the argument that institutional cross-blockholding strengthens investors' coordinating and monitoring power, prior research has documented a positive impact of institutional cross-blockholding on portfolio firms by increasing their market shares [15], improving corporate governance [31], facilitating financial accessibility [32], decreasing financial reporting opacity [33], and enhancing technology spillover [34]. By showing that institutional cross-blockholding decreases firms' CSP through the distraction channel, this article reveals an inadvertent impairment of institutional cross-ownership on the interests of *nonfinancial* stakeholders. Under the prevalence of common ownership [35], policymakers should not overlook the potential cost of institutional cross-ownership through investor distraction. Our article is also related to a broader topic on institutional investors' effect on firms' social engagement [23]. Because of clients' preference for sustainable and responsible investment, institutional investors have to pay more attention to ESG projects (e.g., [12–14, 36]). Our work goes further by showing a reduction in firms' social engagement when investors are distracted due to institutional cross-blockholding.

Second, this article extends the literature on shareholder distraction [37]. Using stock return shocks from other industries to measure shareholder distraction, prior research has documented that shareholder distraction leads to greater managerial opportunism [38], lower diligence of directors [39], and less engagement in CSR [12]. By linking the literature on institutional ownership structure and shareholder distraction, this article complements [18]'s theory on attention allocation under common ownership in terms of portfolio firms' CSP.

Third, this article contributes to CSR literature by providing a novel determinant of CSP (e.g., [11, 40, 41]). Firms' social engagement can be categorized as (1) strategic CSR; (2) not-for-profit CSR; and (3) CSR resulting from agency problems [42–44]. Driven by these different motivations, firms' social performance is determined by firm-level characteristics such as financial conditions, strategic and reputation concerns, managerial opportunism, employee relationship, shareholder engagement, supply-chain relationship, board structure, and CEO

personalities (e.g., [13, 45–50]), external factors such as media and local government (e.g., [8, 51]), industry-level characteristics such as product market competition [52, 53], and macro-level characteristics including political, culture, labor, and legislation systems [44, 54]. Built on the finding that institutional investors increase CSP [12, 55], this article extends academics' and practitioners' understanding of how specific ownership structure affects CSR under the same institutional ownership level.

The remainder of this article is organized as follows. Section 2 describes the sample selection and defines our variables. Section 3 reports the results from a multivariate OLS regression. Section 4 presents the results from a quasi-natural experiment based on mergers between financial institutions. Section 5 provides the results of mechanism tests. Section 6 concludes.

## Sample selection and variable construction

### 1.1 The sample

The sample comes from multiple sources. Firm-level financial data come from Standard & Poor's Compustat-Capital IQ North America Fundamentals Annual (for brevity, Compustat) database. CSP data are from MSCI ESG KLD STATS database (robustness check with Sustainalytics). Institutional holdings data mainly come from Thomson Reuters' Institutional (13F) Holding database. We supplement this database with FactSet's Institutional Holdings database after June 2013 (excluding observations after 2013 does not impact our main results). Analyst coverage data come from Institutional Brokers Estimate System (I\B\E\S) database. EDGAR search volume (ESV) data are from James Ryans' EDGAR Server Log File (Ryans 2017). CEO compensation data are from Standard & Poor's Compustat–Capital IQ Execucomp database [56]. Shareholder proposal data are from Institutional Shareholder Services' (ISS') Shareholder Proposal database.

Observations that satisfy the following criteria were included. (1) Book equity must be positive. (2) Each firm should at least observations for at least two consecutive years. (3) All regressors have values. (4) Firms are not in financial (SIC code 6000–6999) or utility (SIC codes 4900–4999) industries.

The OLS regression uses the estimation sample of 13,112 firm-year observations during 1995–2014. The DID regression uses 3,778 observations during 1995–2012 from 36 effective mergers that are included in the quasi-natural experiment. We alleviate the potential disturbance from outliers by Winsorizing values of all the continuous variables at the $1^{st}$ and the $99^{th}$ percentiles.

### 1.2 Measures of CSP

Firms' CSP is measured by the CSR scores from the KLD database [12, 57, 58]. CSP can reflect in dimensions including community, workforce diversity, employee relations, environment impact, product quality, human rights, corporate governance, and whether firms' business related to alcohol, gaming, firearms, military contracting, nuclear, or tobacco [12, 58, 59]. Therefore, following [12], our CSP measure includes dimensions of community, workforce diversity, employee relations, environment impact, and product quality. We do not consider corporate governance dimension because of (1) our focus on non-financial stakeholders' interests [12], (2) the difference between corporate governance and other issue areas in KLD [49], and (3) doubts on the validity of corporate governance measured in KLD [59].

The CSR score on each dimension is the firm's score on strengths on that dimension minus its score on concerns on that dimension [12]. The overall CSR score is the sum of the CSR scores on the five dimensions. Similarly, the overall CSR strength score (overall CSR concerns score) is the sum of strengths scores (concerns scores) on all dimensions. Lastly, we also use overall CSR score (*CSR*), overall CSR strengths score (*STR*), and overall CSR concerns (*CON*)

score, and five CSR dimension scores—Community (*COM*), Workforce diversity (*DIV*), Employee relations (*EMP*), Environment impact (*ENV*), and Product quality (*PRO*). Table A1 in S1 Appendix defines the variables.

Following He and Huang (2017), the overall CSR score at *t+1* ($CSR_{t+1}$) in the multivariate OLS analysis and the 2-year average overall CSR score ($AvgCSR_{i,[t+1,\ t+2]}$) serve as the outcome measures in the quasi-natural experiment. Similar to prior works [15], we use $AvgCSR_{i,[t+1,\ t+2]}$ in the quasi-natural experiment for the following reasons. First, since CSR is a long-term corporate policy and ESG ratings are not always updated in time [23], the average performance in the following years can provide stronger evidence under the effect of institutional cross-blockholding. Second, due to the concern that more noise will be introduced to CSR measures if a longer period of CSR is implemented. Accordingly, we use a 2-year average of CSP in this analysis as a tradeoff to reflect the effect of institutional cross-blockholding and avoid the potential contamination on CSP by using a longer CSR horizon. Other CSR measures in the main analysis are constructed in a similar way. In the robustness tests, we also use CSR with different horizons as well as CSR score measured by Sustainalytics database as alternative dependent variables.

### 1.3 Measures of institutional cross-blockholding

To measure institutional cross-blockholding, we first extract the quarterly data of institutional investor holdings from Thomson Reuters Institutional (13F) Holdings (adjusted by FactSet Institutional Holdings databases for observations after June 2013). The U.S. Securities and Exchange Commission considers a threshold of 5% stock ownership for any investor to have a material impact in a firm's governance (https://www.sec.gov/news/press-release/2022-22). Therefore, we exclude observations if investors hold shares less than 5% of the firms' total outstanding. Following prior research ([15], He, Huang, and Zhao 2019), we use five measures of institutional cross-blockholding based on Fama-French 48 Industry Classification [60]. *CROSS_DUM* is an indicator that equals 1 if the firm is cross-blockheld by any institutional investors in any quarter in a fiscal year and 0 otherwise. *AVGNUM_Q* is the number of peer firms in the same industry whose stock is held by the same blockholders on a quarterly basis. *CROSS_OWN_Q* is the percentage of shares held by cross-blockholders in each quarter. This measure reflects the total influence that cross-blockholders can exert on firms. Next, we average the quarterly variables *AVGNUM_Q*, *CROSS_OWN_Q* over the fiscal year to have *AVGNUM*, *CROSS_OWN* in each firm-year. *NUMCROSS* is the number of unique cross-blockholders in each firm-year. *NUMCONNECTED* captures the number of firms (i.e., in the same industry) that have any common blockholders with the focal firm in the focal year. Although calculated from different aspects of institutional cross-blockholding, the five measures are highly correlated in an unreported correlation matrix. Following He and Huang (2017), we use *CROSS_DUM* as the independent variable in most of our OLS regressions and include other measures to test robustness.

### 1.4 Control variables

Following prior research [12, 15, 31, 58, 61], we include several control variables to control for firm characteristics that may affect both corporate social responsibility and institutional cross-blockholding (Table A1 in S1 Appendix).

## Multivariate OLS analysis

### 1.5 Summary statistics of multivariate OLS sample

As Table 1 shows, the summary statistics of the variables are similar to those reported in prior research [12, 15].

**Table 1. Summary statistics: Multivariate OLS analysis.** This table reports summary statistics of the key variables in multivariate OLS analysis. The sample comes from multiple sources. Firm-level financial data come from COMPUSTAT database. Corporate social responsibility data come from MSCI ESG KLD database. Institutional investor holdings data come from Thomson Reuters Institutional (13F) Holdings database (adjusted by Factset Institutional Holdings database after June 2013). Analyst coverage data come from Institutional Brokers Estimate System (I\B\E\S). We require observations to satisfy the following criteria: (1) Book equity is positive; (2) Each firm should at least have 2-year consecutive observations; (3) Variables are available in all observations; (4) Firms are not in financial (SIC code 6000–6999) or utility (SIC codes 4900–4999) industries. Finally, the sample consists of 13,112 observations that meet these criteria during 1995–2014 when both Thomson Reuters Institutional (13F) Holdings and KLD are available. All continuous variables are winsorized at $1^{st}$ and $99^{th}$ percentiles to alleviate the potential disturbance from outliers. The variable definitions are provided in Table A1 of S1 Appendix.

| | N | Mean | St.dev | P25 | Median | P75 |
|---|---|---|---|---|---|---|
| *Dependent variables: Corporate social responsibility* | | | | | | |
| CSR | 13,112 | -0.1204 | 1.9680 | -1.0000 | 0.0000 | 1.0000 |
| STR | 13,112 | 1.1268 | 1.8267 | 0.0000 | 0.0000 | 1.0000 |
| CON | 13,112 | 1.2472 | 1.4385 | 0.0000 | 1.0000 | 2.0000 |
| COM | 13,112 | 0.0657 | 0.4647 | 0.0000 | 0.0000 | 0.0000 |
| DIV | 13,112 | -0.0133 | 1.1621 | -1.0000 | 0.0000 | 0.0000 |
| EMP | 13,112 | -0.0718 | 0.8338 | 0.0000 | 0.0000 | 0.0000 |
| ENV | 13,112 | 0.0050 | 0.7250 | 0.0000 | 0.0000 | 0.0000 |
| PRO | 13,112 | -0.1060 | 0.5536 | 0.0000 | 0.0000 | 0.0000 |
| *Independent variables: Institutional cross-blockholding measures* | | | | | | |
| CROSS DUM | 13,112 | 0.7264 | 0.4458 | 0.0000 | 1.0000 | 1.0000 |
| AVGNUM | 13,112 | 2.9864 | 4.4985 | 0.0000 | 1.3750 | 3.5417 |
| CROSS OWN | 13,112 | 0.1068 | 0.0970 | 0.0000 | 0.0903 | 0.1609 |
| NUMCONNECT | 13,112 | 1.9611 | 1.6233 | 0.0000 | 2.0000 | 4.0000 |
| NUMCROSS | 13,112 | 1.6861 | 1.5196 | 0.0000 | 1.0000 | 3.0000 |
| *Control variables* | | | | | | |
| SIZE | 13,112 | 7.1107 | 1.4724 | 6.0258 | 6.9933 | 8.0780 |
| TOBINQ | 13,112 | 2.0195 | 1.2303 | 1.2369 | 1.6212 | 2.3443 |
| BLEV | 13,112 | 0.2183 | 0.1835 | 0.0536 | 0.2009 | 0.3275 |
| EBITDA | 13,112 | 0.1256 | 0.1165 | 0.0837 | 0.1306 | 0.1851 |
| PPENT | 13,112 | 0.2693 | 0.2244 | 0.0955 | 0.1976 | 0.3858 |
| CAPX | 13,112 | 0.0538 | 0.0577 | 0.0194 | 0.0356 | 0.0645 |
| INSTO | 13,112 | 0.7705 | 0.1806 | 0.6623 | 0.8036 | 0.9127 |
| NAN | 13,112 | 2.0806 | 0.8122 | 1.6094 | 2.1972 | 2.7081 |
| RETA | 13,112 | 0.1400 | 0.4051 | 0.0332 | 0.1808 | 0.3493 |
| LN_NUM_INST | 13,112 | 5.1056 | 0.7002 | 4.6299 | 5.0353 | 5.5304 |

Table A2 in S1 Appendix presents the means of institutional cross-blockholding measures by years over the sample period during 1995–2014. As Table A2 in S1 Appendix reports, all measures of institutional cross-blockholding show an increasing trend during the sample period. This trend is consistent with findings in the extant literature [34]. For example, the percentage of cross-held firms rises from 52.44% in 1995 to 81.54% in 2014, and the mean of total cross-ownership increases nearly three times from 0.06 to 0.15. Since the sample only contains big firms covered in the KLD database, the figures on the average U.S. listed firms might be lower, but the trend should hold.

## 1.6 Baseline regressions

Following related literature [12, 15, 58], the following specification examines the relation between institutional cross-blockholding and firms' CSP:

$$CSPMeasure_{i,t+1} = \alpha + \beta CrossMeasure_{i,t} + \gamma Controls_{i,t} + FEs + \varepsilon_{i,t} \qquad (1)$$

where *CSPMeasure* denotes measures on CSP. *CrossMeasure* denotes one of the five proxies on institutional cross-blockholding in each firm-year. In the base regressions, the dependent variable is the overall CSP at $t + 1$ ($CSR_{t+1}$), and the independent variable, *CROSS_DUM*, is an indicator that equals 1 if the firm is cross held in any quarter of the year. The control variables are those described in Section 2.4. *FE* denotes the fixed effects. To control for time-, firm-, industry-, and location-invariant factors as well as all industry-level factors, we include firm- and year-FEs, firm- and industry×year-FEs, as well as firm-, industry×year-, and county-FEs in the regressions. Industries are classified by Fama-French 48 industries [60]. The information of headquarter location (FIPS code) comes from Software Repository for Accounting and Finance (SRAF). Using location information from COMPUSTAT is misleading because COMPUSTAT only gives firms' newest location in its county variable. Standard errors are adjusted for heteroskedasticity and clustered by firm. Table 2 reports the base results. Since location information is missing in some observations, fewer observations are used in regressions with firm, industry×year, and county-fixed effects.

As Table 2 reports, the coefficient estimates of *CROSS_DUM*, ranging from −.1724 to −.1560, are negative and significant at 1% level across alternate model specifications (simple or full model) and different sets of FEs (firm and year; firm and industry×year; and firm, industry×year, and county). In terms of economic magnitude, the average coefficient on *CROSS_DUM* is about −.1643, meaning that firms cross-held by blockholders have .1643 lower scores in future CSP than non-cross-held firms. Given the sample standard deviation of *CSR* (1.9680), the coefficient estimate of *CROSS_DUM* is statistically significant and economically meaningful. Moreover, the coefficient estimates on *CROSS_DUM* are not significantly different when county-FEs are included, meaning that location-invariant factors do not affect the results. To capture firm-, time-, and industry-invariant factors and to alleviate problems mentioned in [61], we implement firm- and industry×year-FEs in the following regressions. The results are similar when we use other sets of fixed effects (firm and year fixed effects and firm, industry×year, and county fixed effects).

Consistent with prior literature [62–64], firms with higher profit (*EBITDA*) and higher retained earnings (*RETA*) perform better on CSR. Further, coefficients on *INSTO* show a significantly positive relation between institutional ownership and CSP, which supports the engaging role of institutional ownership on firms' CSP [12, 14, 36]. The effect of institutional investors on CSR is further supported by the positively significant coefficient on the breadth of institutional ownership (*LN_NUM_INST*) [65]. Overall, this evidence shows that, at the same level of institutional ownership, institutional cross-blockholding is negatively associated with firms' future CSP, which supports the distraction hypothesis.

## 1.7 Robustness tests

First, we use alternate measures of institutional cross-ownership as independent variables. Second, we use CSR measures with different horizons (CSP at *t+2* ($CSR_{t+2}$), 2-year average of CSR ($AvgCSR_{[t+1,t+2]}$), and 3-year average of CSR ($AvgCSR_{[t+1,t+3]}$)) as dependent variables. Third, since the varying number of categories in KLD over the years may result in potential bias, we further use an adjusted CSR score (*AdjCSR*) based on adjusted weights of CSR dimensions as Deng, Kang, and Low (2013) to test if the baseline results are sensitive to the different calculation method of CSR score. Fourth, we exclude observations during the 2008 financial crisis to alleviate the concern that our results are driven by firms' responses to the financial crisis [26]. Fifth, due to the divergence of ESG rating agencies, we use the CSP measured by another ESG rating agency, Sustainalytics database, as our dependent variable to examine whether the baseline results only exist in the KLD data [23–25] (see Table 3).

**Table 2. Baseline regressions: Multivariate OLS analysis.** This table reports the regression results for the association between institutional cross-blockholding and corporate social responsibility. We run the baseline regression as Eq (1):

$$CSRMeasure_{i,t+1} = \alpha + \beta CrossMeasure_{i,t} + \gamma Controls_{i,t} + FE + \varepsilon_{i,t+1} \qquad (1)$$

(1). where *CSRMeasure* is one of the several measures on corporate social responsibility. *CrossMeasure* is one of the five cross-blockholding proxies in each firm-year. In the baseline regressions, the dependent variable is the CSP at *t+1* ($CSR_{t+1}$), and the independent variable, *CROSS_DUM*, is an indicator that equals one if the firm is cross-held in any quarter of the year. The control variables include firm size (*SIZE*), Tobin's Q (*TOBINQ*), book leverage (*BLEV*), profitability (*EBITDA*), collateral (*PPENT*), investment (*CAPX*), institutional ownership (*INSTO*), analyst coverage (*NAN*), retained earnings (*RETA*), and the logarithm form of the number of 13F institutional investors (*LN_NUM_INST*). Three sets of fixed effects are included in the regressions. Columns (1) and (2) report results with firm and year fixed effects. Columns (3) and (4) report the results of models with firm and industry×year fixed effects. Columns (5) and (6) report results with firm, industry×year, and county fixed effects. Results of the simple model are provided in Columns (1), (3), and (5), and the results of the full model are shown in Columns (2), (4), and (6). Industries are classified by Fama-French 48 industries [60]. Standard errors are adjusted for heteroskedasticity and clustered by firm. *, **, and *** indicate significance at the 10%, 5%, and 1% levels, respectively. Standard errors are shown in the parentheses.

| | (1) | (2) | (3) | (4) | (5) | (6) |
|---|---|---|---|---|---|---|
| | $CSR_{t+1}$ | $CSR_{t+1}$ | $CSR_{t+1}$ | $CSR_{t+1}$ | $CSR_{t+1}$ | $CSR_{t+1}$ |
| CROSS_DUM | -0.1600*** | -0.1680*** | -0.1571*** | -0.1720*** | -0.1560*** | -0.1724*** |
| | (0.0478) | (0.0473) | (0.0474) | (0.0476) | (0.0496) | (0.0498) |
| SIZE | | -0.1559* | | -0.1279 | | -0.1341* |
| | | (0.0799) | | (0.0779) | | (0.0803) |
| TOBINQ | | -0.0571** | | -0.0385 | | -0.0396 |
| | | (0.0251) | | (0.0248) | | (0.0260) |
| BLEV | | 0.3842* | | 0.1585 | | 0.1941 |
| | | (0.1970) | | (0.1929) | | (0.2004) |
| EBITDA | | 0.4818** | | 0.3349 | | 0.2456 |
| | | (0.2118) | | (0.2109) | | (0.2117) |
| PPENT | | 0.6873 | | 0.4770 | | 0.3999 |
| | | (0.4326) | | (0.3966) | | (0.4111) |
| CAPX | | -1.1595** | | -0.8646* | | -0.8733 |
| | | (0.4853) | | (0.5188) | | (0.5553) |
| INSTO | | 0.1265 | | 0.3943** | | 0.4062** |
| | | (0.2065) | | (0.1975) | | (0.2060) |
| NAN | | 0.0404 | | 0.0029 | | 0.0089 |
| | | (0.0381) | | (0.0374) | | (0.0391) |
| RETA | | 0.1801* | | 0.1629* | | 0.1813* |
| | | (0.1001) | | (0.0950) | | (0.0971) |
| LN_NUM_INST | | 0.4148*** | | 0.3558*** | | 0.3425*** |
| | | (0.1222) | | (0.1158) | | (0.1213) |
| Constant | -0.0042 | -1.3664** | -0.0063 | -1.3161** | 0.0044 | -1.1917** |
| | (0.0347) | (0.6000) | (0.0344) | (0.5845) | (0.0362) | (0.6000) |
| Firm FE | Yes | Yes | Yes | Yes | Yes | Yes |
| Year FE | Yes | Yes | No | No | No | No |
| Industry×Year FE | No | No | Yes | Yes | Yes | Yes |
| County FE | No | No | No | No | Yes | Yes |
| Observations | 13,112 | 13,112 | 13,112 | 13,112 | 12,530 | 12,530 |
| R-squared | 0.6564 | 0.6588 | 0.6998 | 0.7016 | 0.6929 | 0.6946 |

**1.7.1 Alternate measures of institutional cross-ownership.** Following [15], we use alternate measures of institutional cross-ownership as independent variables. These measures include: (1) the average number of peers (i.e., firms in the focal firm's industry) whose stock are held by the same blockholders (*AVGNUM*); (2) the percentage of shares held by cross-blockholders (*CROSS_OWN*); (3) the number of unique cross-blockholders (*NUMCROSS*); and

**Table 3. Robustness tests of the OLS analysis.** This table presents the results of the robustness tests of the OLS analysis. Panel A reports the results when alternative measures of institutional cross-blockholding are used as independent variables. The alternative measures of institutional cross-blockholding include the average number of peers in the same industry that are held by the same blockholders (*AVGNUM*), the total percentage of shares held by cross-blockholders (*CROSS_OWN*), the number of unique cross-blockholders (*NUMCROSS*), the number of peer firms in the same industry that share any common blockholder with the firm (*NUMCONNECT*). Panel B shows the results when applying CSP with different horizons. CSP at *t+2* ($CSR_{t+2}$), 2-year average of CSR ($AvgCSR_{[t+1,t+2]}$), and 3-year average of CSR ($AvgCSR_{[t+1,t+3]}$) are used in this analysis. Panel C reports results when the dependent variable is the adjusted CSP ($AdjCSR_{t+1}$) calculated according to Deng, Kang, and Low (2013). Panel D shows the results when excluding observations during the 2008 financial crisis. Panel E shows the results when firms' CSP is rated by Sustainalytics database. *SOSICALS* and *AvgSOCIALS* and denote the social score and 2-year average social score in Sustainalytics, respectively. The control variables are the same as those in baseline regressions. Industries are classified by Fama-French 48 industries. Standard errors are adjusted for heteroskedasticity and clustered by firm. *, **, and *** indicate significance at the 10%, 5%, and 1% levels, respectively. Standard errors are shown in the parentheses.

**Panel A: Alternative measures of institutional cross-blockholding**

|  | (1) | (2) | (3) | (4) | (5) | (6) | (7) | (8) |
|---|---|---|---|---|---|---|---|---|
|  | $CSR_{t+1}$ | $CSR_{t+1}$ | $CSR_{t+1}$ | $CSR_{t+1}$ | $CSR_{t+1}$ | $CSR_{t+1}$ | $CSR_{t+1}$ | $CSR_{t+1}$ |
| *AVGNUM* | -0.0249*** | -0.0184*** |  |  |  |  |  |  |
|  | (0.0067) | (0.0063) |  |  |  |  |  |  |
| *CROSS_OWN* |  |  | -0.4827* | -0.4448* |  |  |  |  |
|  |  |  | (0.2581) | (0.2597) |  |  |  |  |
| *NUMCROSS* |  |  |  |  | -0.0472*** | -0.0463*** |  |  |
|  |  |  |  |  | (0.0177) | (0.0178) |  |  |
| *NUMCONNECT* |  |  |  |  |  |  | -0.0289** | -0.0263** |
|  |  |  |  |  |  |  | (0.0122) | (0.0124) |
| Constant | -1.4085** | -1.5024** | -1.3670** | -1.4291** | -1.3839** | -1.4469** | -1.3750** | -1.4296** |
|  | (0.5850) | (0.6025) | (0.5884) | (0.6062) | (0.5875) | (0.6041) | (0.5875) | (0.6032) |
| Controls | Yes | Yes | Yes | Yes | Yes | Yes | Yes | Yes |
| Firm FE | Yes | Yes | Yes | Yes | Yes | Yes | Yes | Yes |
| Year FE | No | Yes | No | Yes | No | Yes | No | Yes |
| Industry×Year FE | Yes | No | Yes | No | Yes | No | Yes | No |
| Observations | 13,112 | 13,112 | 13,112 | 13,112 | 13,112 | 13,112 | 13,112 | 13,112 |
| R-squared | 0.7017 | 0.6586 | 0.7012 | 0.6583 | 0.7014 | 0.6585 | 0.7012 | 0.6584 |

**Panel B: Alternative CSR horizons**

|  | (1) | (2) | (3) | (4) | (5) | (6) |  |  |
|---|---|---|---|---|---|---|---|---|
|  | $CSR_{t+2}$ | $CSR_{t+2}$ | $AvgCSR_{[t+1, t+2]}$ | $AvgCSR_{[t+1, t+2]}$ | $AvgCSR_{[t+1, t+3]}$ | $AvgCSR_{[t+1, t+3]}$ |  |  |
| *CROSS_DUM* | -0.1509*** | -0.1651*** | -0.1540*** | -0.1686*** | -0.1138** | -0.1240*** |  |  |
|  | (0.0501) | (0.0503) | (0.0452) | (0.0454) | (0.0452) | (0.0453) |  |  |
| Constant | 0.0522 | -1.3078** | 0.0229 | -1.3119** | 0.0151 | -0.9646 |  |  |
|  | (0.0364) | (0.5893) | (0.0329) | (0.5693) | (0.0324) | (0.5978) |  |  |
| Controls | No | Yes | No | Yes | No | Yes |  |  |
| Firm FE | Yes | Yes | Yes | Yes | Yes | Yes |  |  |
| Industry ×Year FE | Yes | Yes | Yes | Yes | Yes | Yes |  |  |
| Observations | 13,112 | 13,112 | 13,112 | 13,112 | 11,076 | 11,076 |  |  |
| R-squared | 0.6957 | 0.6979 | 0.7492 | 0.7512 | 0.7941 | 0.7959 |  |  |

**Panel C: Alternative calculation method of CSR**

|  | (1) | (2) | (3) | (4) | (5) | (6) |  |  |
|---|---|---|---|---|---|---|---|---|
|  | $AdjCSR_{t+1}$ | $AdjCSR_{t+1}$ | $AdjCSR_{t+1}$ | $AdjCSR_{t+1}$ | $AdjCSR_{t+1}$ | $AdjCSR_{t+1}$ |  |  |
| *CROSS_DUM* | -0.0305*** | -0.0335*** | -0.0355*** | -0.0370*** | -0.0298*** | -0.0330*** |  |  |
|  | (0.0091) | (0.0091) | (0.0095) | (0.0093) | (0.0095) | (0.0095) |  |  |
| Constant | -0.0722*** | -0.1227 | -0.0686*** | -0.1358 | -0.0707*** | -0.0898 |  |  |
|  | (0.0066) | (0.1138) | (0.0069) | (0.1138) | (0.0069) | (0.1169) |  |  |
| Controls | No | Yes | No | Yes | No | Yes |  |  |
| Firm FE | Yes | Yes | Yes | Yes | Yes | Yes |  |  |
| Year FE | No | No | Yes | Yes | No | No |  |  |
| Industry ×Year FE | Yes | Yes | No | No | Yes | Yes |  |  |

*(Continued)*

**Table 3.** (Continued)

| | | | | | | |
|---|---|---|---|---|---|---|
| County FE | No | No | No | No | Yes | Yes |
| Observations | 13,112 | 13,112 | 13,112 | 13,112 | 12,530 | 12,530 |
| R-squared | 0.6760 | 0.6776 | 0.6310 | 0.6334 | 0.6671 | 0.6688 |

**Panel D: Excluding observations in the 2008 financial crisis**

| | (1) | (2) | (3) | (4) | (5) | (6) |
|---|---|---|---|---|---|---|
| | $CSR_{t+1}$ | $CSR_{t+1}$ | $CSR_{t+1}$ | $CSR_{t+1}$ | $CSR_{t+1}$ | $CSR_{t+1}$ |
| CROSS_DUM | -0.1514*** | -0.1639*** | -0.1618*** | -0.1677*** | -0.1537*** | -0.1672*** |
| | (0.0535) | (0.0541) | (0.0539) | (0.0537) | (0.0563) | (0.0568) |
| Constant | 0.0803** | -1.4026** | 0.0879** | -1.4022** | 0.0898** | -1.2780** |
| | (0.0386) | (0.6072) | (0.0390) | (0.6297) | (0.0409) | (0.6238) |
| Controls | No | Yes | No | Yes | No | Yes |
| Firm FE | Yes | Yes | Yes | Yes | Yes | Yes |
| Year FE | No | No | Yes | Yes | No | No |
| Industry×Year FE | Yes | Yes | No | No | Yes | Yes |
| County FE | No | No | No | No | Yes | Yes |
| Observations | 10,890 | 10,890 | 10,890 | 10,890 | 10,414 | 10,414 |
| R-squared | 0.7088 | 0.7106 | 0.6655 | 0.6682 | 0.7009 | 0.7027 |

**Panel E: CSR measures from Sustainalytics sample**

| | (1) | (2) | (3) | (4) | (5) | (6) |
|---|---|---|---|---|---|---|
| | $SOCIALS_{t+1}$ | $SOCIALS_{t+1}$ | $SOCIALS_{t+1}$ | $SOCIALS_{t+1}$ | $SOCIALS_{t+1}$ | $AvgSOCIALS$ |
| CROSS_DUM | -0.8395*** | | | | | -0.7966*** |
| | (0.3147) | | | | | (0.3041) |
| AVGNUM | | -0.0847** | | | | |
| | | (0.0340) | | | | |
| CROSS_OWN | | | -2.9870** | | | |
| | | | (1.5076) | | | |
| NUMCROSS | | | | -0.1774** | | |
| | | | | (0.0873) | | |
| NUMCONNECT | | | | | -0.1369** | |
| | | | | | (0.0677) | |
| Constant | -4.7713 | -5.4566 | -4.9304 | -5.3399 | -5.2195 | 2.1382 |
| | (6.2675) | (6.1682) | (6.2371) | (6.2141) | (6.1921) | (7.4716) |
| Controls | Yes | Yes | Yes | Yes | Yes | Yes |
| Firm FE | Yes | Yes | Yes | Yes | Yes | Yes |
| Industry×Year FE | Yes | Yes | Yes | Yes | Yes | Yes |
| Observations | 2,144 | 2,144 | 2,144 | 2,144 | 2,144 | 2,144 |
| R-squared | 0.9564 | 0.9564 | 0.9562 | 0.9562 | 0.9563 | 0.9595 |

(4) the number of peer firms that have at least one blockholder that the focal firm also has (NUMCONNECT).

Panel A of Table 3 reports that all the coefficients of interest are negative and highly significant when using alternate measures of institutional cross-blockholding in both simple and full models. Different from CROSS_DUM, the alternate measures, which are all continuous, provide more information on the economic significance of institutional cross-blockholding on firms' CSP. For example, Column (1) shows that a one-standard-deviation increase in AVGNUM is associated with a .02 × 4.49 = .09 decrease in CSR. Compared to the sample mean of CSR, this evidence shows that the coefficient on AVGNUM is significant statistically and economically. Other measures in Panel A show similar results.

**1.7.2 Different horizons of CSR.** Further, four variables examine whether the base results are sensitive to the horizon of CSR: (1) CSP in $t + 2$ ($CSR_{t+2}$); (2) two-year average of CSP from $t + 1$ to $t + 2$ ($AvgCSR_{[t+1,t+2]}$); and (4) three-year average of CSP from $t + 1$ to $t + 3$ ($AvgCSR_{[t+1,t+3]}$). Panel B of Table 3 reports the results. Like the base results, the coefficient estimates of interest are negative and highly significant across all the dependent measures, meaning that base results are not sensitive to the horizon of CSP.

**1.7.3 Alternate calculation.** Our base results may be sensitive to how one calculates CSR score. To alleviate this concern, we re-calculate the CSR score based on the adjusted weight [61] and create *AdjCSR* as our dependent measure. Panel C of Table 3 reports that the coefficients of interest are negative and significant in all regressions. Compared to the mean *AdjCSR* of $-.0830$, the coefficients of interest are significant statistically and economically. We thus conclude that the calculation methods of CSR do not affect the base results.

**1.7.4 Excluding observations in financial crisis.** To address the concern that the effect of institutional cross-blockholding on firms' CSP is caused by firms' different responses to the financial crisis [26], we exclude observations during the 2008 financial crises to avoid the potential intervention of financial crisis. Excluding those observations reduces the sample size to 10,890. Panel D of Table 3 shows that the coefficients of interest are negative and significant across different specifications. In terms of magnitude, the coefficients of interest are close to those estimated in the base regressions. We conclude that the financial crisis likely does not drive our results.

**1.7.5 CSP measure from Sustainalytics.** CSR scores vary by rating agencies [24, 25]. Our results may thus be sensitive to our use of KLD database. We test the sensitivity by using CSP from Sustainalytics database. While the KLD database focuses on U.S. listed firms, Sustainalytics database covers global firms, but for a shorter set of years than KLD. Due to the difference in firm coverage, only 2,144 observations in our sample are linked to the Sustainalytics database. In spite of limited observations, we still find a negative and significant relation between institutional cross-blockholding and firms' CSP. Panel E of Table 3 shows the results when CSR is measured as the Social Score in Sustainalytics database. This evidence is robust to alternate measures of institutional cross-blockholding (Columns 1 through 5) and the measure of CSP (Column 7).

## Identification: A quasi-natural experiment

### 1.8 Mergers between financial institutions

As [15] mentioned, the majority of mergers between financial institutions come from financial regulations (e.g., the Riegle–Neal Interstate Banking and Branching Efficiency Act of 1994, and the Financial Services Modernization Act of 1999). Therefore, mergers between institutional blockholders provide an ideal quasi-natural experiment because the mergers (1) are less likely to be affected by their portfolio firms' decisions (relevance condition), and (2) can affect only portfolio firms through affected institutional investors (exclusion restriction). Following [15], we start from merger information and blockholders' holding information in the Securities Data Company (SDC) database and Thomson Reuters Institutional (13F) Holdings database. According to [15], we require mergers to satisfy the following four criteria. (1) Both acquirer and target in the merger must be located in the United States and recorded in the institutional holdings sample. (2) The adjusted institutional holdings sample stops recording the filings of the target institutional blockholders in the same year as the merger's announcement date. (3) The completion period of mergers should not exceed one year. (4) The target and acquirer held firms in overlapped industry at one quarter before the merger. After identifying the list of mergers, we link affected portfolio firms to financial data from COMPUSTAT.

**Table 4. Summary statistics: Quasi-natural experiment.** This table reports the summary statistics of the key variables in the quasi-natural experiment based on mergers between financial institutional blockholders during 1995–2012. The sample comes from multiple sources. Firm-level financial data come from COMPUSTAT database. Corporate social responsibility data come from MSCI ESG KLD database. Institutional investor holdings data come from Thomson Reuters Institutional (13F) Holdings database. Analyst coverage data come from Institutional Brokers Estimate System (I\B\E\S). We require observations to satisfy the following criteria: (1) Book equity is positive; (2) Each firm should at least have 2-year consecutive observations; (3) Variables are available in all observations; (4) Firms are not in financial (SIC code 6000–6999) or utility (SIC codes 4900–4999) industries. Our sample includes 3,778 firm-years that meet these criteria during 1995–2012 when Thomson Reuters Institutional (13F) Holdings and KLD are available and firms can be matched to blockholder mergers. All continuous variables are winsorized at $1^{st}$ and $99^{th}$ percentiles to alleviate the potential disturbance from outliers.

| | N | Mean | Std. Dev. | P25 | Median | P75 |
|---|---|---|---|---|---|---|
| *Dependent variables*: *Corporate social responsibility* | | | | | | |
| $AvgCSR_{[t+1, t+2]}$ | 3,778 | -0.3677 | 1.9675 | -1.5000 | -0.5000 | 0.5000 |
| $AvgSTR_{[t+1, t+2]}$ | 3,778 | 1.0776 | 1.7890 | 0.0000 | 0.5000 | 1.0000 |
| $AvgCON_{[t+1, t+2]}$ | 3,778 | 1.4452 | 1.3669 | 0.5000 | 1.0000 | 2.0000 |
| $AvgCOM_{[t+1, t+2]}$ | 3,778 | 0.0719 | 0.4551 | 0.0000 | 0.0000 | 0.0000 |
| $AvgDIV_{[t+1, t+2]}$ | 3,778 | -0.1102 | 1.2293 | -1.0000 | 0.0000 | 0.5000 |
| $AvgEMP_{[t+1, t+2]}$ | 3,778 | -0.1784 | 0.7823 | -0.5000 | 0.0000 | 0.0000 |
| $AvgENV_{[t+1, t+2]}$ | 3,778 | -0.0318 | 0.6786 | 0.0000 | 0.0000 | 0.0000 |
| $AvgPRO_{[t+1, t+2]}$ | 3,778 | -0.1191 | 0.5004 | 0.0000 | 0.0000 | 0.0000 |
| *Control variables* | | | | | | |
| SIZE | 3,778 | 6.8182 | 1.3273 | 5.8754 | 6.6533 | 7.5830 |
| TOBINQ | 3,778 | 1.8989 | 1.1008 | 1.2028 | 1.5570 | 2.2017 |
| BLEV | 3,778 | 0.2004 | 0.1798 | 0.0238 | 0.1817 | 0.3097 |
| EBITDA | 3,778 | 0.1234 | 0.1214 | 0.0817 | 0.1311 | 0.1850 |
| PPENT | 3,778 | 0.2569 | 0.2155 | 0.0910 | 0.1925 | 0.3667 |
| CAPX | 3,778 | 0.0510 | 0.0552 | 0.0184 | 0.0337 | 0.0614 |
| INSTO | 3,778 | 0.8221 | 0.1519 | 0.7351 | 0.8513 | 0.9400 |
| NAN | 3,778 | 1.9962 | 0.7194 | 1.6094 | 2.0794 | 2.4849 |
| RETA | 3,778 | 0.1619 | 0.4208 | 0.0648 | 0.2083 | 0.3858 |
| LN NUM INST | 3,778 | 4.9935 | 0.5614 | 4.6299 | 4.9318 | 5.2946 |

To make the analysis comparable to related research on CSR, firms in financial (SIC code: 6000–6999) and utility (SIC code: 4900–4999) industries are excluded. Observations with missing data are excluded as well. If no firms are from a merger because of missing variables, the merger will be dropped out of our sample. Portfolio firms that become cross-held after mergers are identified as treated firms. Other portfolio firms of related blockholders are identified as control firms. Finally, 3,778 observations during 1995–2012 from 36 effective mergers are included in the quasi-natural experiment.

Table 4 reports the summary statistics of variables this analysis uses. Fig 1 shows the number of effective mergers between institutional blockholders each year during 1995–2012.

Fig 1 differs from the corresponding figure in He and Huang (2017) for several reasons. First, we exclude firms in financial/utility industries whereas [15] keep those observations. Second, because Thomson Reuters Institutional (13F) Holdings database contains some errors [22], the database needs to delete or correct some information on institutional investors over time. This change makes some blockholders mentioned in [15] no longer available in our sample. Fortunately, the majority of [15]'s cases in 1995–2010 stay, making our analysis consistent with prior research.

Following [15], we use the 2-year average CSP $AvgCSR_{[t+1, t+2]}$ as the dependent variable and implement a symmetric 7-year window around the event year.

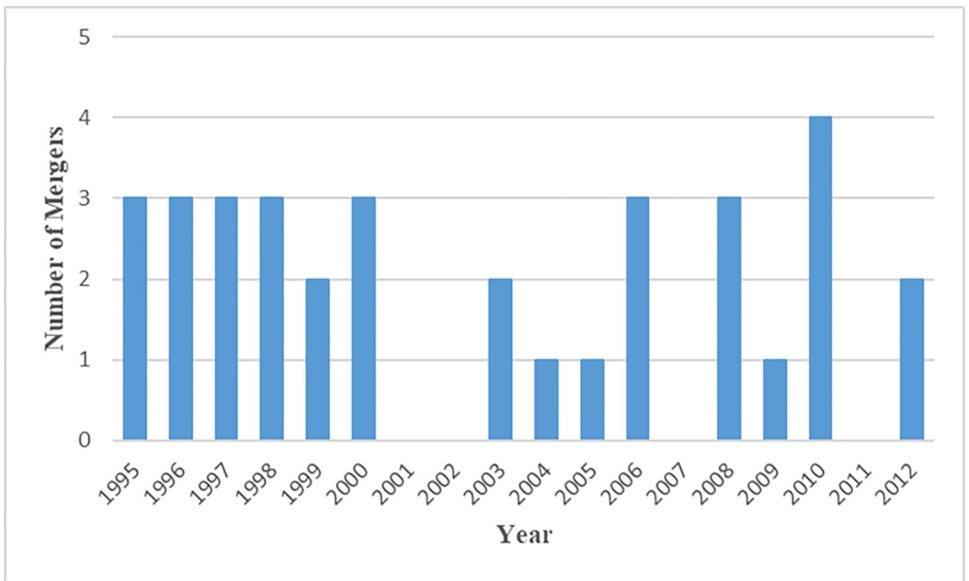

**Fig 1. Number of mergers between institutional blockholders.** This figure plots the number of effective mergers between institutional blockholders in each year during 1995–2012. The merger information and blockholders' holding information come from SDC database and Thomson Reuters Institutional (13F) Holdings database, respectively. According to [15], mergers should satisfy the following criteria. First, both acquirer and target in the merger are located in the U.S. and should be listed in Thomson Reuters Institutional (13F) Holdings as institutional blockholders that hold more than 5% of shares in at least one firm. Second, Thomson Reuters Institutional (13F) Holdings database stops recording the fillings of the target institutional blockholders in the same year as that of the merger's announcement date. Third, the completion period of mergers should not exceed one year. Fourth, both the target and acquirer hold firms in the same industry at one quarter before the merger. After that, we link firms affected by these mergers to financial data from COMPUSTAT. In order to make the analysis comparable to related works on CSR, firms in financial (SIC code: 6000–6999) and utility (SIC code: 4900–4999) industries are excluded. Observations with missing data are also excluded. If no firms are from a merger case because they are from financial and utility industries or with missing data, the merger will be dropped out of the merger cases. Finally, 36 effective merger cases are included in this quasi-natural experiment.

## 1.9 The DID results

Under this quasi-natural experiment, we estimate the average treatment effect (ATE) of the exogenous shocks by running the following regression:

$$AvgCSRMeasure_{[t+1,t+2]}$$
$$= \alpha + \beta_1 Treat_i \times Post_t + \beta_2 Treat_i + \beta_3 Post_t + \gamma' Controls_{i,t} + FE + \varepsilon_{i,t+1} \quad (3)$$

where $AvgCSRMeasure_{[t+1, t+2]}$ denotes the two-year average CSP measures. Control variables are the same as those in base regressions. *FE* denotes firm- and merger-FEs and firm×merger FEs, as [15] suggested. Standard errors are adjusted for heteroskedasticity and clustered by firm.

Panel A of Table 5 reports that all the coefficients of interest, *TREAT×POST*, are negative and significant at 1% level across model specifications and sets of FEs. The ATE effect is about −.3853 across different settings. The interpretation is that cross-blockheld firms, on average, reduce CSP by .3853 each year in the following two years. Compared to the sample mean and standard deviation of overcall CSR score, this effect is statistically significant and economically meaningful. Consistent with estimates from the OLS regression, this evidence provides a causal inference on the negative impact of institutional cross-blockholding on firms' CSP, which corroborates the distraction hypothesis.

**Table 5. Quasi-natural experiment: Institutional blockholder mergers.** This table reports the results of the difference-in-differences tests under a quasi-natural experiment based on mergers between institutional blockholders located in the U.S. during 1995–2012. Following [15], we apply a symmetric 7-year window around the event year. Panel A shows the results in the unmatched sample. Panel B shows the DID results after a propensity score matching, where the matching variables include firm size, Tobin'Q, book leverage, collateral, investment, profitability, analyst coverage, and institutional ownership, retained earnings, breadth of institutional ownership, industry dummies, and year dummies. The dependent variable, $AvgCSR_{[t+1, t+2]}$, is the 2-year average of firms' CSP. Standard errors shown in the parentheses are adjusted for heteroskedasticity and clustered by firm. *, **, and *** indicate significance at the 10%, 5%, and 1% levels, respectively.

**Panel A: Unmatched sample**

|  | (1) | (2) | (3) | (4) | (5) | (6) |
|---|---|---|---|---|---|---|
|  | $AvgCSR_{[t+1, t+2]}$ | $AvgCSR_{[t+1, t+2]}$ | $AvgCSR_{[t+1, t+2]}$ | $AvgCSR_{[t+1, t+2]}$ | $AvgCSR_{[t+1, t+2]}$ | $AvgCSR_{[t+1, t+2]}$ |
| TREAT× POST | -0.4471*** | -0.4154*** | -0.3252*** | -0.4227*** | -0.3874*** | -0.3142*** |
|  | (0.1354) | (0.1240) | (0.1226) | (0.1336) | (0.1242) | (0.1204) |
| TREAT | 0.1669 | 0.1120* |  | 0.1650 | 0.0823 |  |
|  | (0.1184) | (0.0607) |  | (0.1150) | (0.0653) |  |
| POST | 0.1532** | 0.1908*** | 0.1806*** | 0.1530** | 0.1735*** | 0.1689*** |
|  | (0.0688) | (0.0605) | (0.0589) | (0.0705) | (0.0614) | (0.0606) |
| Constant | -0.4156*** | -0.4226*** | -0.4000*** | -3.3513*** | -0.9338 | -1.9470* |
|  | (0.0737) | (0.0220) | (0.0217) | (0.8319) | (1.0183) | (1.0393) |
| Controls | No | No | No | Yes | Yes | Yes |
| Firm×Merger FE | No | No | Yes | No | No | Yes |
| Firm FE | No | Yes | No | No | Yes | No |
| Merger FE | No | Yes | No | No | Yes | No |
| Observations | 3,778 | 3,720 | 3,475 | 3,778 | 3,720 | 3,475 |
| R-squared | 0.0020 | 0.8008 | 0.8268 | 0.0992 | 0.8030 | 0.8291 |

**Panel B: Matched sample**

|  | (1) | (2) | (3) | (4) | (5) | (6) |
|---|---|---|---|---|---|---|
|  | $AvgCSR_{[t+1, t+2]}$ | $AvgCSR_{[t+1, t+2]}$ | $AvgCSR_{[t+1, t+2]}$ | $AvgCSR_{[t+1, t+2]}$ | $AvgCSR_{[t+1, t+2]}$ | $AvgCSR_{[t+1, t+2]}$ |
| TREAT× POST | -0.3118* | -0.4668** | -0.3321* | -0.4449** | -0.4449** | -0.3166* |
|  | (0.1835) | (0.1814) | (0.1765) | (0.1794) | (0.1794) | (0.1756) |
| TREAT | 0.0055 | 0.1706 |  | 0.1739 | 0.1739 |  |
|  | (0.1695) | (0.1201) |  | (0.1216) | (0.1216) |  |
| POST | 0.0146 | 0.2540* | 0.1560 | 0.2557* | 0.2557* | 0.1520 |
|  | (0.1415) | (0.1356) | (0.1303) | (0.1415) | (0.1415) | (0.1357) |
| Constant | -0.2508* | -0.3551*** | -0.2563*** | -0.0053 | -0.0053 | -1.7868 |
|  | (0.1319) | (0.0676) | (0.0410) | (1.8547) | (1.8547) | (1.7402) |
| Controls | No | No | No | Yes | Yes | Yes |
| Firm×Merger FE | No | No | Yes | No | No | Yes |
| Firm FE | No | Yes | No | No | Yes | No |
| Merger FE | No | Yes | No | No | Yes | No |
| Observations | 1,238 | 1,042 | 937 | 1,042 | 1,042 | 937 |
| R-squared | 0.0033 | 0.8017 | 0.8295 | 0.8062 | 0.8062 | 0.8342 |

## 1.10 Robustness tests

**1.10.1 Propensity score matching.** We use the nearest-neighbor matching of propensity scores. The matching variables include firm size, Tobin's Q, book leverage, collateral, investment, analyst coverage, institutional ownership, retained earnings, breadth of institutional ownership, industry dummies, and year dummies. Each treated firm is matched to two firms in the control group. Finally, the matched sample consists of 1,434 observations. To test whether this matching satisfies the balance condition, we adopt a balance test of PSM and find that firms' characteristics are balanced between treatment and control groups (Table A3 in S1 Appendix). After PSM, we replicate the DID analysis and show the results in Panel B of

**Table 6. PSM-DID: Alternative matching variables.** This table presents the robustness test of PSM-DID by using alternative sets of matching variables. Only coefficients of interest (*TREAT*×*POST*) are included in this table. Columns (1) and (2) show the average treatment effects of regressions with firm and merger fixed effects. Columns (3) and (4) report the average treatment effects of regression with firm×merger fixed effect. Standard errors are adjusted for heteroskedasticity and clustered by firm. *, **, and *** indicate significance at the 10%, 5%, and 1% levels, respectively. Standard errors are shown in the parentheses.

| | (1) | (2) | (3) | (4) |
|---|---|---|---|---|
| | $AvgCSR_{[t+1,\,t+2]}$ | $AvgCSR_{[t+1,\,t+2]}$ | $AvgCSR_{[t+1,\,t+2]}$ | $AvgCSR_{[t+1,\,t+2]}$ |
| **Panel A: Propensity score matching on *SIZE*** | | | | |
| *TREAT*× *POST* | -0.5591*** | -0.5412*** | -0.4609** | -0.4457** |
| | (0.1915) | (0.1900) | (0.1990) | (0.1987) |
| **Panel B: Propensity score matching on *SIZE/TOBINQ*** | | | | |
| *TREAT*× *POST* | -0.4847** | -0.4794** | -0.4108** | -0.4106** |
| | (0.1875) | (0.1870) | (0.1936) | (0.1953) |
| **Panel C: Propensity score matching on *SIZE/TOBINQ/INSTO*** | | | | |
| *TREAT*× *POST* | -0.4901*** | -0.5023*** | -0.3941** | -0.3973** |
| | (0.1863) | (0.1863) | (0.1893) | (0.1904) |
| **Panel D: Propensity score matching on *SIZE/TOBINQ/RETA*** | | | | |
| *TREAT*× *POST* | -0.4869*** | -0.4795** | -0.4076** | -0.4027** |
| | (0.1877) | (0.1866) | (0.1939) | (0.1950) |
| **Panel E: Propensity score matching on *SIZE/TOBINQ/EBITDA*** | | | | |
| *TREAT*× *POST* | -0.4726** | -0.4736** | -0.3812** | -0.3897** |
| | (0.1871) | (0.1852) | (0.1913) | (0.1905) |
| **Panel F: Propensity score matching on *SIZE/TOBINQ/RETA/INSTO*** | | | | |
| *TREAT*× *POST* | -0.4935*** | -0.5049*** | -0.3941** | -0.3973** |
| | (0.1864) | (0.1864) | (0.1893) | (0.1904) |
| **Panel G: Propensity score matching on *SIZE/TOBINQ/EBITDA/INSTO*** | | | | |
| *TREAT*× *POST* | -0.4973*** | -0.4842** | -0.3924** | -0.3798* |
| | (0.1897) | (0.1872) | (0.1937) | (0.1933) |
| Controls | No | Yes | No | Yes |
| Firm×Merger FE | No | No | Yes | Yes |
| Firm FE | Yes | Yes | No | No |
| Merger FE | Yes | Yes | No | No |

Table 5. Like the results in Panel A of Table 5, the coefficients of interest are all negative and significant in the matched sample. This evidence further supports the negative effect of institutional cross-blockholding on firms' CSP.

To see whether the results in Panel B of Table 5 are sensitive to the selection of matching variables, we use alternate sets of matching variables in PSM. In these robustness tests, we use the following sets of matching variables: (1) *SIZE*; (2) *SIZE/TOBINQ*; (3) *SIZE/TOBINQ/INSTO*; (4) *SIZE/TOBINQ/RETA*; (5) *SIZE/TOBINQ/EBITDA*; (6) *SIZE/TOBINQ/RETA/INSTO*; and (7) *SIZE/TOBINQ/EBITDA/INSTO*. Table 6 reports the ATEs of institutional cross-blockholding are negative and significant across different model specifications. In addition, the magnitude of the treatment effect varies little across different matching variables and model settings. Overall, these results suggest that the negative impact of institutional cross-blockholding on firms' CSP is not driven by the selection of matching variables.

**1.10.2 Excluding the merger between BlackRock and Barclays Global Investors.** As [26] showed, the merger between BlackRock and Barclays Global Investors contributes a large portion of observations in the quasi-natural experiment among institutional blockholders mergers. Because the BlackRock-BGI merger is a sensation under the background of the financial crisis, portfolio firms' action to the merger may reflect firms' response to other factors

**Table 7. Excluding the merger between BlackRock and Barclays Global Investors.** This table presents the effect of institutional cross-blockholding on corporate social responsibility when excluding the merger between BlackRock and Barclays Global Investors. The dependent variable is $AvgCSR_{[t+1, t+2]}$. All the model specifications are the same as those in Table 5. Columns (1) and (4) show results without any fixed effects. Columns (2) and (5) show results with merger fixed effect. Columns (3) and (6) report results with firm×merger fixed effect. Results without control variables are presented in Columns (1)-(3). Columns (4)-(5) show results with control variables as those in baseline regressions. Standard errors are adjusted for heteroskedasticity and clustered by firm. *, **, and *** indicate significance at the 10%, 5%, and 1% levels, respectively. Standard errors are shown in the parentheses.

| | (1) | (2) | (3) | (4) | (5) | (6) |
|---|---|---|---|---|---|---|
| | $AvgCSR_{[t+1, t+2]}$ | $AvgCSR_{[t+1, t+2]}$ | $AvgCSR_{[t+1, t+2]}$ | $AvgCSR_{[t+1, t+2]}$ | $AvgCSR_{[t+1, t+2]}$ | $AvgCSR_{[t+1, t+2]}$ |
| TREAT× POST | -0.6028*** | -0.4475*** | -0.3550** | -0.5808*** | -0.4099*** | -0.3411** |
| | (0.1675) | (0.1519) | (0.1414) | (0.1669) | (0.1565) | (0.1387) |
| TREAT | 0.0558 | 0.0958 | | 0.1860 | 0.0299 | |
| | (0.1330) | (0.0608) | | (0.1325) | (0.0701) | |
| POST | 0.2150** | 0.2069** | 0.1680** | 0.2070* | 0.1751* | 0.1515* |
| | (0.1092) | (0.0881) | (0.0828) | (0.1108) | (0.0901) | (0.0865) |
| Constant | -0.2194** | -0.1982*** | -0.1586*** | -2.4049** | -0.7927 | -1.7171 |
| | (0.0984) | (0.0339) | (0.0307) | (1.0714) | (1.6661) | (1.4870) |
| Controls | No | No | No | Yes | Yes | Yes |
| Firm×Merger FE | No | No | Yes | No | No | Yes |
| Firm FE | No | Yes | No | No | Yes | No |
| Merger FE | No | Yes | No | No | Yes | No |
| Observations | 1,790 | 1,602 | 1,543 | 1,790 | 1,602 | 1,543 |
| R-squared | 0.0059 | 0.8238 | 0.8461 | 0.1002 | 0.8272 | 0.8483 |

(e.g., financial crisis) instead of institutional cross-ownership [26]. Therefore, we examine the change of firms' CSP after blockholder mergers excluding observations affected by the BlackRock-BGI merger. Table 7 shows the results.

Similar to [26]'s findings, the sample size drops to 1,790 after excluding observations related to the BlackRock-BGI merger. However, our DID results are significant in this restricted sample. We conclude that the impact of institutional cross-blockholding on CSR comes not from firms' response to the financial crisis.

## 1.11 Decomposition of CSR: Strengths and concerns

As Table 8 shows, the average treatment effects on CSR concerns are positive and significant in all the models, while the effects on CSR strengths are significant only in Columns (1) and (4) of Panel A. This evidence suggests that institutional cross-blockholding reduces firms' CSP mainly through increasing CSR concerns. The insight is that investors are more likely to overlook firms' CSR concerns when they are distracted under institutional cross-blockholding [12].

## 1.12 CSR dimensions

Following Chen, Dong, and Lin (2020), we include five CSR dimensions in this analysis, including Community (*COM*), Workforce diversity (*DIV*), Employee relations (*EMP*), Environment impact (*ENV*), and Product quality (*PRO*). The construction of dependent variables, control variables, and fixed effects are the same as those in Table 4. Table 9 reports the results.

As Table 9 reports, the impact of institutional cross-blockholding is significant on workforce diversity, employee relations, and product quality dimensions but insignificant on community and environment dimensions. Because investors punish firms severely if their performance on community and diversity dimensions drops [59], our results suggest the stickiness of firm performance on these two CSR dimensions.

**Table 8. Strengths and concerns.** This table presents the effect of institutional cross-blockholding on firms' performance on CSR strengths and concerns in the difference-in-differences analysis. Panel A shows the effect of institutional cross-blockholding on CSR strengths. Panel B shows the effect of institutional cross-blockholding on CSR concerns. Columns (1) and (4) show results without any fixed effects. Columns (2) and (5) show results with merger fixed effect. Columns (3) and (6) report results with firm×merger fixed effect. Results without control variables are presented in Columns (1)-(3). Columns (4)-(5) show results with control variables as those in baseline regressions. Standard errors are adjusted for heteroskedasticity and clustered by firm. *, **, and *** indicate significance at the 10%, 5%, and 1% levels, respectively. Standard errors are shown in the parentheses.

**Panel A: Strengths**

|  | (1) | (2) | (3) | (4) | (5) | (6) |
|---|---|---|---|---|---|---|
|  | $AvgSTR_{[t+1,\ t+2]}$ | $AvgSTR_{[t+1,\ t+2]}$ | $AvgSTR_{[t+1,\ t+2]}$ | $AvgSTR_{[t+1,\ t+2]}$ | $AvgSTR_{[t+1,\ t+2]}$ | $AvgSTR_{[t+1,\ t+2]}$ |
| TREAT× POST | -0.1727* | -0.1034 | -0.0483 | -0.1658* | -0.0857 | -0.0549 |
|  | (0.1043) | (0.0898) | (0.0886) | (0.0970) | (0.0899) | (0.0878) |
| TREAT | 0.0828 | 0.0949** |  | 0.0345 | 0.0583 |  |
|  | (0.1050) | (0.0411) |  | (0.0850) | (0.0434) |  |
| POST | -0.0256 | -0.0250 | -0.0273 | -0.0916* | -0.0565 | -0.0494 |
|  | (0.0563) | (0.0455) | (0.0442) | (0.0544) | (0.0458) | (0.0454) |
| Constant | 1.0871*** | 1.0814*** | 1.1134*** | -5.2060*** | -0.4648 | -0.7684 |
|  | (0.0669) | (0.0165) | (0.0163) | (0.7179) | (0.8269) | (0.8626) |
| Controls | No | No | No | Yes | Yes | Yes |
| Firm×Merger FE | No | No | Yes | No | No | Yes |
| Firm FE | No | Yes | No | No | Yes | No |
| Merger FE | No | Yes | No | No | Yes | No |
| Observations | 3,778 | 3,720 | 3,475 | 3,778 | 3,720 | 3,475 |
| R-squared | 0.0004 | 0.8693 | 0.8882 | 0.3413 | 0.8703 | 0.8892 |

**Panel B: Concerns**

|  | (1) | (2) | (3) | (4) | (5) | (6) |
|---|---|---|---|---|---|---|
|  | $AvgCON_{[t+1,\ t+2]}$ | $AvgCON_{[t+1,\ t+2]}$ | $AvgCON_{[t+1,\ t+2]}$ | $AvgCON_{[t+1,\ t+2]}$ | $AvgCON_{[t+1,\ t+2]}$ | $AvgCON_{[t+1,\ t+2]}$ |
| TREAT× POST | 0.2744*** | 0.3120*** | 0.2769*** | 0.2568*** | 0.3017*** | 0.2593*** |
|  | (0.0866) | (0.0778) | (0.0739) | (0.0846) | (0.0779) | (0.0739) |
| TREAT | -0.0841 | -0.0171 |  | -0.1305* | -0.0240 |  |
|  | (0.0965) | (0.0429) |  | (0.0772) | (0.0473) |  |
| POST | -0.1788*** | -0.2158*** | -0.2079*** | -0.2446*** | -0.2300*** | -0.2183*** |
|  | (0.0449) | (0.0409) | (0.0402) | (0.0452) | (0.0427) | (0.0407) |
| Constant | 1.5027*** | 1.5040*** | 1.5134*** | -1.8547*** | 0.4690 | 1.1786* |
|  | (0.0537) | (0.0149) | (0.0142) | (0.4166) | (0.6743) | (0.6590) |
| Controls | No | No | No | Yes | Yes | Yes |
| Firm×Merger FE | No | No | Yes | No | No | Yes |
| Firm FE | No | Yes | No | No | Yes | No |
| Merger FE | No | Yes | No | No | Yes | No |
| Observations | 3,778 | 3,720 | 3,475 | 3,778 | 3,720 | 3,475 |
| R-squared | 0.0036 | 0.8163 | 0.8448 | 0.2181 | 0.8188 | 0.8471 |

## Mechanism tests

### 1.13 Testing the distraction channel

**1.13.1 Evidence from EDGAR search volume.** To find direct evidence on how investor attention changes, we use EDGAR search volume (*ESV*) measured by [39] to measure attention [28, 29, 39, 66] and examine whether institutional cross-blockholding leads to investor distraction [12, 38, 67]. All the ESV measures are in the logarithm form. We use EDGAR search volume from James Ryans' EDGAR Log File Data (http://www.jamesryans.com/). Different from the shareholder distraction measure in [38], EDGAR search volume directly

**Table 9. Institutional cross-blockholding and performance in CSR dimensions.** This table reports the effect of institutional cross-blockholding on firms' performance in different CSR dimensions. CSR dimensions include Community (*COM*), Workforce diversity (*DIV*), Employee relations (*EMP*), Environment impact (*ENV*), and Product quality (*PRO*). The dependent variable is the 2-year average of CSR score in each dimension. Results for CSP in Community, Workforce diversity, Employee relations, Environment impact, and Product quality dimensions are shown in Panels A, B, C, D, and E, respectively. Columns (1) and (4) show results without any fixed effects. Columns (2) and (5) show results with merger fixed effect. Columns (3) and (6) report results with firm×merger fixed effect. Results without control variables are presented in Columns (1)-(3). Columns (4)-(5) show results with control variables as those in baseline regressions. Standard errors are adjusted for heteroskedasticity and clustered by firm. *, **, and *** indicate significance at the 10%, 5%, and 1% levels, respectively. Standard errors are shown in the parentheses.

**Panel A: Community**

|  | (1) | (2) | (3) | (4) | (5) | (6) |
|---|---|---|---|---|---|---|
|  | $AvgCOM_{[t+1, t+2]}$ | $AvgCOM_{[t+1, t+2]}$ | $AvgCOM_{[t+1, t+2]}$ | $AvgCOM_{[t+1, t+2]}$ | $AvgCOM_{[t+1, t+2]}$ | $AvgCOM_{[t+1, t+2]}$ |
| TREAT× POST | 0.0039 | 0.0168 | 0.0223 | 0.0057 | 0.0182 | 0.0242 |
|  | (0.0307) | (0.0275) | (0.0276) | (0.0309) | (0.0277) | (0.0276) |
| TREAT | -0.0252 | -0.0101 |  | -0.0240 | -0.0128 |  |
|  | (0.0266) | (0.0136) |  | (0.0256) | (0.0140) |  |
| POST | 0.0649*** | 0.0598*** | 0.0626*** | 0.0567*** | 0.0532*** | 0.0595*** |
|  | (0.0165) | (0.0151) | (0.0151) | (0.0175) | (0.0156) | (0.0153) |
| Constant | 0.0516*** | 0.0514*** | 0.0516*** | -0.3591 | 0.5085* | 0.4482 |
|  | (0.0161) | (0.0054) | (0.0055) | (0.2212) | (0.2836) | (0.3134) |
| Controls | No | No | No | Yes | Yes | Yes |
| Firm×Merger FE | No | No | Yes | No | No | Yes |
| Firm FE | No | Yes | No | No | Yes | No |
| Merger FE | No | Yes | No | No | Yes | No |
| Observations | 3,778 | 3,720 | 3,475 | 3,778 | 3,720 | 3,475 |
| R-squared | 0.0048 | 0.7506 | 0.7908 | 0.0638 | 0.7530 | 0.7923 |

**Panel B: Workforce diversity**

|  | (1) | (2) | (3) | (4) | (5) | (6) |
|---|---|---|---|---|---|---|
|  | $AvgDIV_{[t+1, t+2]}$ | $AvgDIV_{[t+1, t+2]}$ | $AvgDIV_{[t+1, t+2]}$ | $AvgDIV_{[t+1, t+2]}$ | $AvgDIV_{[t+1, t+2]}$ | $AvgDIV_{[t+1, t+2]}$ |
| TREAT× POST | -0.2222*** | -0.1898*** | -0.1458** | -0.2151*** | -0.1777** | -0.1429* |
|  | (0.0806) | (0.0724) | (0.0729) | (0.0780) | (0.0721) | (0.0730) |
| TREAT | 0.1431* | 0.0564 |  | 0.1302* | 0.0395 |  |
|  | (0.0730) | (0.0368) |  | (0.0696) | (0.0399) |  |
| POST | -0.3774*** | -0.3835*** | -0.3837*** | -0.3804*** | -0.3843*** | -0.3744*** |
|  | (0.0397) | (0.0332) | (0.0331) | (0.0387) | (0.0337) | (0.0340) |
| Constant | 0.0213 | 0.0311** | 0.0408*** | -2.9050*** | -1.8950*** | -2.1314*** |
|  | (0.0456) | (0.0121) | (0.0123) | (0.4607) | (0.5867) | (0.5873) |
| Controls | No | No | No | Yes | Yes | Yes |
| Firm×Merger FE | No | No | Yes | No | No | Yes |
| Firm FE | No | Yes | No | No | Yes | No |
| Merger FE | No | Yes | No | No | Yes | No |
| Observations | 3,778 | 3,720 | 3,475 | 3,778 | 3,720 | 3,475 |
| R-squared | 0.0261 | 0.8240 | 0.8463 | 0.2251 | 0.8288 | 0.8516 |

**Panel C: Employee relations**

|  | (1) | (2) | (3) | (4) | (5) | (6) |
|---|---|---|---|---|---|---|
|  | $AvgEMP_{[t+1, t+2]}$ | $AvgEMP_{[t+1, t+2]}$ | $AvgEMP_{[t+1, t+2]}$ | $AvgEMP_{[t+1, t+2]}$ | $AvgEMP_{[t+1, t+2]}$ | $AvgEMP_{[t+1, t+2]}$ |
| TREAT× POST | -0.1106* | -0.1342** | -0.1268** | -0.1052* | -0.1283** | -0.1224** |
|  | (0.0609) | (0.0587) | (0.0549) | (0.0609) | (0.0582) | (0.0542) |
| TREAT | 0.1220** | 0.0212 |  | 0.1217** | 0.0194 |  |
|  | (0.0568) | (0.0316) |  | (0.0557) | (0.0328) |  |
| POST | 0.1673*** | 0.1909*** | 0.1875*** | 0.1682*** | 0.1800*** | 0.1774*** |
|  | (0.0285) | (0.0292) | (0.0297) | (0.0298) | (0.0303) | (0.0304) |
| Constant | -0.2475*** | -0.2419*** | -0.2330*** | -0.4460 | -0.0464 | 0.0513 |

(*Continued*)

**Table 9.** (Continued)

| | (0.0295) | (0.0103) | (0.0106) | (0.2851) | (0.5052) | (0.5170) |
|---|---|---|---|---|---|---|
| Controls | No | No | No | Yes | Yes | Yes |
| Firm×Merger FE | No | No | Yes | No | No | Yes |
| Firm FE | No | Yes | No | No | Yes | No |
| Merger FE | No | Yes | No | No | Yes | No |
| Observations | 3,778 | 3,720 | 3,475 | 3,778 | 3,720 | 3,475 |
| R-squared | 0.0108 | 0.7218 | 0.7574 | 0.0323 | 0.7244 | 0.7604 |
| **Panel D: Environment impact** | | | | | | |
| | (1) | (2) | (3) | (4) | (5) | (6) |
| | $AvgENV_{[t+1,\,t+2]}$ | $AvgENV_{[t+1,\,t+2]}$ | $AvgENV_{[t+1,\,t+2]}$ | $AvgENV_{[t+1,\,t+2]}$ | $AvgENV_{[t+1,\,t+2]}$ | $AvgENV_{[t+1,\,t+2]}$ |
| *TREAT× POST* | -0.0449 | -0.0273 | 0.0094 | -0.0357 | -0.0252 | 0.0019 |
| | (0.0473) | (0.0448) | (0.0445) | (0.0469) | (0.0444) | (0.0438) |
| *TREAT* | -0.0940* | 0.0434** | | -0.0870* | 0.0354 | |
| | (0.0490) | (0.0220) | | (0.0450) | (0.0233) | |
| *POST* | 0.2048*** | 0.2263*** | 0.2175*** | 0.2101*** | 0.2207*** | 0.2038*** |
| | (0.0269) | (0.0283) | (0.0273) | (0.0281) | (0.0299) | (0.0279) |
| Constant | -0.0901*** | -0.1150*** | -0.1090*** | -0.4266* | 0.4607 | -0.2382 |
| | (0.0241) | (0.0105) | (0.0095) | (0.2468) | (0.4174) | (0.4555) |
| Controls | No | No | No | Yes | Yes | Yes |
| Firm×Merger FE | No | No | Yes | No | No | Yes |
| Firm FE | No | Yes | No | No | Yes | No |
| Merger FE | No | Yes | No | No | Yes | No |
| Observations | 3,778 | 3,720 | 3,475 | 3,778 | 3,720 | 3,475 |
| R-squared | 0.0212 | 0.7134 | 0.7444 | 0.0679 | 0.7181 | 0.7491 |
| **Panel E: Product quality** | | | | | | |
| | (1) | (2) | (3) | (4) | (5) | (6) |
| | $AvgPRO_{[t+1,\,t+2]}$ | $AvgPRO_{[t+1,\,t+2]}$ | $AvgPRO_{[t+1,\,t+2]}$ | $AvgPRO_{[t+1,\,t+2]}$ | $AvgPRO_{[t+1,\,t+2]}$ | $AvgPRO_{[t+1,\,t+2]}$ |
| *TREAT× POST* | -0.0732** | -0.0810** | -0.0842*** | -0.0724** | -0.0745** | -0.0749** |
| | (0.0333) | (0.0333) | (0.0317) | (0.0330) | (0.0338) | (0.0313) |
| *TREAT* | 0.0211 | 0.0012 | | 0.0242 | 0.0008 | |
| | (0.0323) | (0.0163) | | (0.0299) | (0.0179) | |
| *POST* | 0.0936*** | 0.0972*** | 0.0967*** | 0.0983*** | 0.1039*** | 0.1027*** |
| | (0.0192) | (0.0177) | (0.0175) | (0.0198) | (0.0177) | (0.0174) |
| Constant | -0.1508*** | -0.1482*** | -0.1505*** | 0.7855*** | 0.0384 | -0.0769 |
| | (0.0188) | (0.0063) | (0.0063) | (0.1931) | (0.2711) | (0.2749) |
| Controls | No | No | No | Yes | Yes | Yes |
| Firm×Merger FE | No | No | Yes | No | No | Yes |
| Firm FE | No | Yes | No | No | Yes | No |
| Merger FE | No | Yes | No | No | Yes | No |
| Observations | 3,778 | 3,720 | 3,475 | 3,778 | 3,720 | 3,475 |
| R-squared | 0.0069 | 0.7529 | 0.7858 | 0.1183 | 0.7552 | 0.7887 |

measures investor attention from the demand side of information. According to [39], the majority of information acquisition of EDGAR comes from sophisticated investors. We use the logarithm form of the total number of nonrobot page views according to [39]'s method (*Total ESV*) as the dependent variable. We also decompose *Total ESV* by the types of SEC filings. Following Iliev et al. (2021), we separate SEC filings into financial and nonfinancial filings. Financial filings (*ESV Financial*) include 10-K and 10-Q filings, and others are identified

as non-financial filings (*ESV Non-financial*). To eliminate the intervention of attention from government sectors, we exclude attention from IRS [68] and construct the total search volume of nonrobot page viewers excluding IRS search records (*Non-IRS ESV*) as another dependent variable.

Panel A of Table 10 reports the coefficient estimates of *TREAT×POST* are significantly negative for EDGAR search volume in total filings, financial filings, nonfinancial filings, and total ESV excluding IRS search records. The findings support the distraction effect of institutional cross-blockholding. In addition, the attention to nonfinancial filings decreases more than the attention to financial filings. This finding is consistent with our prediction that distracted investors may focus less on corporate information that is less related to their portfolio performance.

If the distraction hypothesis is correct, the negative impact of institutional cross-blockholding on CSP is expected to be higher for firms with lower market attention before the mergers because managers in those firms face less pressure from investors' engagement in CSR due to distraction [12].

As Panel B of Table 10 shows, the effect of institutional cross-blockholding on CSP significant for firms with low ESV but insignificant for firms with high ESV. This evidence supports our prediction on investor distraction because firms with lower attention before mergers are more likely to be overlooked when investors become busier after the merger. To test the external validity of these results, we conduct similar tests on the multivariate OLS sample and find similar results (Table A4 in S1 Appendix). One possible caveat of this analysis is that we do not track investors' EDGAR search volume one by one around mergers. However, because the decrease in attention via EDGAR is less likely driven by other market participants around the mergers, our results should be stronger when we remove such a noise from overall EDGAR attention.

Evidence from different types of institutional investors also supports the distraction hypothesis. Using Brian Bushee's classification [69, 70] to identify the type of institutional investors, we observe a significant decrease in CSR for firms cross-held by transient investors and quasi-indexer but not by dedicated investors (Table A5 in S1 Appendix). Because dedicated investors pay more attention to their portfolio firms, these results are consistent with the distraction thesis.

**1.13.2 Evidence from shareholder proposals.** If blockholders are distracted by the increased portfolio firms, we would observe a decrease in shareholder proposals [18]. Following [12], we collect shareholder proposals on socially responsible investment from the ISS Shareholder Proposals database. The dependent variables $\%SRI_{t+1}$, $\%SRI\_PASS_{t+1}$, $NUM\_SRI_{t+1}$, and $NUM\_SRI\_PASS_{t+1}$ denote percent of proposals on SRI, percent of passed proposals on SRI, the number of proposals on SRI, and the number of passed proposals on SRI, respectively. According to the distraction hypothesis, if institutional cross-blockholders pay less attention to firms' CSR activities, we will observe fewer SRI shareholder proposals for firms under institutional cross-blockholding.

Panel A of Table 11 shows that the percentage of SRI proposals and passed SRI proposals are lower in firms under institutional cross-blockholding. Panel B of Table 11 reports similar results in terms of the number of SRI proposals. Because shareholder proposals reflect the intention of shareholders [12, 30], the decrease in SRI proposals provides direct evidence on how much attention institutional cross-blockholder pay to firms' CSP. Based on findings from EDGAR attention and shareholder proposals, we infer that the negative impact of institutional cross-blockholding on portfolio firms' CSP comes mainly from investors' distraction when holding multiple firms simultaneously.

**Table 10. Distraction channel: Evidence from EDGAR search volume.** This table shows the impact of institutional cross-blockholding on CSR through investor distraction measured by EDGAR search volume. Panel A shows the change of EDGAR search volume (ESV) after blockholder mergers. The ESV is calculated according to [50]. *Total ESV*, *ESV Financial*, *ESV Non-financial* denote the total EDGAR search volume, financial filings' search volume, and non-financial filings' search volume, respectively. *Non-IRS ESV* denotes the total search volume from non-robot viewers excluding IRS, where IRS search data come from [68]. Panel B reports the cross-sectional effects of cross-blockholding on CSR across the level of investor attention before blockholder mergers. A firm is assigned to the High (Low) group if the value of the attention measure is above (below) the median of the sample. A Wald test is implemented to test the difference of ATEs between the High and the Low groups. Firm and merger fixed effects are included in the models. Control variables are the same as those in the baseline regressions. Standard errors are clustered by firm. *, **, and *** indicate significance at the 10%, 5%, and 1% levels, respectively. Standard errors are shown in the parentheses.

**Panel A: Institutional cross-blockholding and investor attention**

| | (1) | (2) | (3) | (4) |
|---|---|---|---|---|
| | *Total ESV* | *ESV Financial* | *ESV Non-financial* | *Non-IRS ESV* |
| TREAT×POST | -0.1151*** | -0.1154*** | -0.1288*** | -0.1155*** |
| | (0.0338) | (0.0345) | (0.0372) | (0.0338) |
| TREAT | 0.3028*** | 0.2992*** | 0.3085*** | 0.3030*** |
| | (0.0260) | (0.0273) | (0.0309) | (0.0261) |
| POST | 0.6625*** | 0.7304*** | 0.5968*** | 0.6622*** |
| | (0.0188) | (0.0204) | (0.0205) | (0.0188) |
| Constant | 4.4782*** | 3.0098*** | 3.9322*** | 4.4696*** |
| | (0.3878) | (0.4542) | (0.4505) | (0.3882) |
| Controls | Yes | Yes | Yes | Yes |
| Firm FE | Yes | Yes | Yes | Yes |
| Merger FE | Yes | Yes | Yes | Yes |
| Observations | 3,396 | 3,396 | 3,396 | 3,396 |
| R-squared | 0.8330 | 0.8407 | 0.7908 | 0.8329 |

**Panel B: Cross-sectional effects across investor attention**

| | (1) | (2) | (3) | (4) |
|---|---|---|---|---|
| | $AvgCSR_{[t+1, t+2]}$ | $AvgCSR_{[t+1, t+2]}$ | $AvgCSR_{[t+1, t+2]}$ | $AvgCSR_{[t+1, t+2]}$ |
| TREAT× POST×High Total ESV | -0.1819 | | | |
| | (0.2229) | | | |
| TREAT× POST×Low Total ESV | -0.5602*** | | | |
| | (0.1338) | | | |
| TREAT× POST×High ESV Financial | | -0.1572 | | |
| | | (0.2098) | | |
| TREAT× POST×Low ESV Financial | | -0.6041*** | | |
| | | (0.1368) | | |
| TREAT× POST×High Non-financial ESV | | | -0.1313 | |
| | | | (0.2277) | |
| TREAT× POST×Low Non-financial ESV | | | -0.5819*** | |
| | | | (0.1333) | |
| TREAT× POST×High Non-IRS ESV | | | | -0.1819 |
| | | | | (0.2229) |
| TREAT× POST×Low Non-IRS ESV | | | | -0.5602*** |
| | | | | (0.1338) |
| Controls, Firm FE, Merger FE | Yes | Yes | Yes | Yes |
| Observations | 3,396 | 3,396 | 3,396 | 3,396 |
| R-squared | 0.7975 | 0.7976 | 0.7976 | 0.7975 |
| Difference: High-:Low | 0.3783* | 0.4468** | 0.4505* | 0.3783* |
| p-value of Wald test | [0.0985] | [0.0401] | [0.0531] | [0.0985] |

**Table 11. Distraction channel: Evidence from shareholder proposals.** This table reports the impact of institutional cross-blockholding on firms' proposals on socially responsible investment (SRI). The percentage and number of all SRI proposals and passed SRI proposals are used in this analysis. The dependent variables $\%SRI_{t+1}$, $\%SRI\_PASS_{t+1}$, $NUM\_SRI_{t+1}$, and $NUM\_SRI\_PASS_{t+1}$ denote the percent of proposals on SRI, the percent of passed proposals on SRI, the number of proposals on SRI, and the number of passed proposals on SRI, respectively. Panel A shows the results for the percentage of SRI proposals, and Panel B reports the results for the number of SRI proposals. The shareholder proposals data come from the ISS Shareholder Proposals database. The independent variable and control variables are the same as those in the baseline regressions. Firm and industry×year fixed effects are included to control for the invariant factors on firm and all factors at the industry level. Industries are classified by Fama-French 48 Industries. Standard errors are adjusted for heteroskedasticity and clustered by firm. *, **, and *** indicate significance at the 10%, 5%, and 1% levels, respectively. Standard errors are shown in the parentheses. For brevity, we only report coefficients of interest and constant terms.

**Panel A: Percentage of proposals on SRI**

|  | (1) | (2) | (3) | (4) |
|---|---|---|---|---|
|  | $\%SRI_{t+1}$ | $\%SRI_{t+1}$ | $\%SRI\_PASS_{t+1}$ | $\%SRI\_PASS_{t+1}$ |
| CROSS_DUM | -0.0216*** | -0.0210*** | -0.0042* | -0.0045* |
|  | (0.0081) | (0.0080) | (0.0022) | (0.0023) |
| Constant | 0.0729*** | 0.0770 | 0.0046*** | 0.0021 |
|  | (0.0058) | (0.0890) | (0.0016) | (0.0166) |
| Controls | No | Yes | No | Yes |
| Firm FE | Yes | Yes | Yes | Yes |
| Industry×Year FE | Yes | Yes | Yes | Yes |
| Observations | 5,893 | 5,893 | 5,893 | 5,893 |
| R-squared | 0.4459 | 0.4468 | 0.2162 | 0.2182 |

**Panel B: Number of proposals on SRI**

|  | (1) | (2) | (3) | (4) |
|---|---|---|---|---|
| Variables | $NUM\_SRI_{t+1}$ | $NUM\_SRI_{t+1}$ | $NUM\_SRI\_PASS_{t+1}$ | $NUM\_SRI\_PASS_{t+1}$ |
| CROSS_DUM | -0.0169* | -0.0158* | -0.0032* | -0.0033* |
|  | (0.0091) | (0.0089) | (0.0017) | (0.0017) |
| Constant | 0.0864*** | 0.1111 | 0.0039*** | -0.0035 |
|  | (0.0066) | (0.1088) | (0.0012) | (0.0214) |
| Controls | No | Yes | No | Yes |
| Firm FE | Yes | Yes | Yes | Yes |
| Industry×Year FE | Yes | Yes | Yes | Yes |
| Observations | 5,893 | 5,893 | 5,893 | 5,893 |
| R-squared | 0.5467 | 0.5473 | 0.2183 | 0.2193 |

## Conclusion

This paper documents the negative impact of institutional cross-blockholding on portfolio firms' CSP. In multivariate OLS regressions, we observe a negative effect of institutional cross-blockholding on firms' CSP, which is robust to alternative measures on CSR and institutional cross-blockholding, different model settings, different sample coverage, different sample period and another source of CSP from Sustainalytics. Using a quasi-natural experiment based on mergers between financial institutions, we establish a causal link between institutional cross-blockholding and portfolio firms' CSP. Examining CSR strengths and concerns, we find an asymmetric impact of institutional cross-blockholding on firms' CSP, which comes more likely from investor distraction according to prior literature. In terms of different CSR dimensions, cross-held firms perform worse in workforce diversity, employee relations, and product quality dimensions, but not in community and environment impact dimensions, which suggests the different social costs of institutional cross-blockholding across CSR dimensions.

Based on more direct evidence from EDGAR search volume and shareholder proposals on socially responsible investment, we infer that the negative effect of institutional cross-blockholding on CSR is more likely to be driven by the distraction effect when institutional investors hold multiple firms at the same time. In heterogeneity tests by corporate governance

environment and product market competition, we eliminate alternative explanations on corporate governance and anticompetitive effects. Overall, by providing evidence from firms' CSP under institutional cross-blockholding, this paper supports [18]'s theory on the allocation of limited attention when investors hold multiple blocks. By documenting an inadvertent impairment of institutional cross-blockholding on firms' CSP due to investor distraction, this paper also suggests another channel through which common investors may affect portfolio firms' behavior. This argument helps us to better understand the impact of common ownership, especially when its anticompetitive effect is questioned by recent research [26, 71]. Public policymakers, proponents of CSR, and financial regulators (e.g., the U.S. Securities and Exchange Commission) may use our findings and caution socially oriented investors from the potential perils of becoming too fragmented in their portfolio. Because such investors aim to use their investments to promote socially beneficial outcomes, they would achieve their aims more effectively and efficiently by focusing their attention on a more coherent set of stocks.

## Supporting information

**S1 Appendix.**
(DOCX)

## Author Contributions

**Conceptualization:** Tao Chen.

**Data curation:** Jimmy Chengyuan Qu.

**Formal analysis:** Jimmy Chengyuan Qu.

**Investigation:** Jimmy Chengyuan Qu.

**Methodology:** Tao Chen.

**Project administration:** Vivek Astvansh.

**Supervision:** Vivek Astvansh.

**Writing – original draft:** Jimmy Chengyuan Qu.

**Writing – review & editing:** Vivek Astvansh, Jimmy Chengyuan Qu.

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
