## [Decision Letter · Decision Letter 0]

18 Jan 2023

PONE-D-22-35109The Social Cost of Investor Distraction: Evidence from Institutional Cross-BlockholdingPLOS ONE

Dear Dr. Vivek Astvansh,

Thank you for submitting your manuscript to PLOS ONE. After careful consideration, we feel that it has merit but does not fully meet PLOS ONE’s publication criteria as it currently stands. Therefore, we invite you to submit a revised version of the manuscript that addresses the points raised during the review process. After careful reading of the manuscript, i agree with the reviewers on the below issues raised1. use of they and we is not appropirate for academic writing.2. professional editing is needed.3. author need to look at the model specification and reduce the number of robust tests.

We look forward to receiving your revised manuscript.

Kind regards,

Kofi Mintah Oware, Ph.D

Academic Editor

PLOS ONE

3. We note that Figure 1 in your submission contain copyrighted images. All PLOS content is published under the Creative Commons Attribution License (CC BY 4.0), which means that the manuscript, images, and Supporting Information files will be freely available online, and any third party is permitted to access, download, copy, distribute, and use these materials in any way, even commercially, with proper attribution. For more information, see our copyright guidelines: http://journals.plos.org/plosone/s/licenses-and-copyright.

Reviewers' comments:

Reviewer's Responses to Questions

**Comments to the Author**

1. Is the manuscript technically sound, and do the data support the conclusions?

Reviewer #1: Yes

Reviewer #2: Yes

2. Has the statistical analysis been performed appropriately and rigorously? 

Reviewer #1: Yes

Reviewer #2: Yes

3. Have the authors made all data underlying the findings in their manuscript fully available?

Reviewer #1: No

Reviewer #2: Yes

4. Is the manuscript presented in an intelligible fashion and written in standard English?

Reviewer #1: Yes

Reviewer #2: Yes

5. Review Comments to the Author

Reviewer #1: I suggest the use of models in research should be reflective of adequately, practically and competently measuring the constructs in the research. So even though the models used here are scientifically rigorous I suggest a non-linear model would have been more adequate, practical and competent.

Reviewer #2: REVIEW REPORT

Manuscript Number: PONE-D-22-35109

Title: The Social Cost of Investor Distraction: Evidence from Institutional Cross-Blockholding

Comments: The study objective is clear but the manuscript in its current form is not suitable for publication in PLOS ONE.

I, therefore recommend: Minor Revision

1. Abstract section need to be revised especially the use of the word ‘They’.

2. Is the benefits of CSR only less tangible to investors in short term? What about the long-run effects/benefits of CSR.

3. Suggest that this statement read as ‘In this test, we first find that firms under greater institutional cross-blockholding receive less attention from investors…………..’

4. Authors should try as much as possible to avoid the use of the word ‘we’. Example can be found in this statement ‘We require observations to satisfy the following criteria:…….’

5. ‘Finally, we use overall CSR score (CSR), overall CSR strengths and concerns scores (STR and CON),……’I suggest that authors revise the statement especially to make it clear as to what STR and CON represent respectively in the statement.

6. Authors should provide the justification for the exclusion of observations if investors hold shares less than 5% of the firms’ total outstanding.

7. Authors need to look at the model specification (i.e. eq. 1) again.

8. Authors have done so many robustness test which distort the flow on what the study seek to achieve. I suggest authors perform only robustness test on the key study objective.

9. Authors should provide strong policy implications of the study.

10. An editing help from someone with full professional proficiency in English is strongly recommended.

6. PLOS authors have the option to publish the peer review history of their article (what does this mean?). If published, this will include your full peer review and any attached files.

Reviewer #1: **Yes: **Halidu Babamu Osman

Reviewer #2: **Yes: **Kingsley Appiah

---

## [Author Response · Author response to Decision Letter 0]

14 Mar 2023

Please refer to the uploaded "Response to Reviewers.docx" file.

---

## [Decision Letter · Decision Letter 1]

15 May 2023

The Social Cost of Investor Distraction: Evidence from Institutional Cross-Blockholding

PONE-D-22-35109R1

Dear Dr. Astvansh,

We’re pleased to inform you that your manuscript has been judged scientifically suitable for publication and will be formally accepted for publication once it meets all outstanding technical requirements.

Kind regards,

Kofi Mintah Oware, Ph.D

Academic Editor

PLOS ONE

Additional Editor Comments (optional):

Reviewers' comments:

Reviewer's Responses to Questions

**Comments to the Author**

1. If the authors have adequately addressed your comments raised in a previous round of review and you feel that this manuscript is now acceptable for publication, you may indicate that here to bypass the “Comments to the Author” section, enter your conflict of interest statement in the “Confidential to Editor” section, and submit your "Accept" recommendation.

Reviewer #1: All comments have been addressed

Reviewer #3: All comments have been addressed

2. Is the manuscript technically sound, and do the data support the conclusions?

Reviewer #1: Yes

Reviewer #3: Yes

3. Has the statistical analysis been performed appropriately and rigorously? 

Reviewer #1: Yes

Reviewer #3: Yes

4. Have the authors made all data underlying the findings in their manuscript fully available?

Reviewer #1: Yes

Reviewer #3: Yes

5. Is the manuscript presented in an intelligible fashion and written in standard English?

Reviewer #1: Yes

Reviewer #3: Yes

6. Review Comments to the Author

Reviewer #1: Topic and issues are interesting.

Literature review was also adequate.

The methodology is rigorous.

The analysis is also appropriate.

Policy recommendation also appropriate.

Reviewer #3: There are a number of grammatical errors and formatting to be done. For example, 'Our theories premise is as follow.' should have a ':' or ';' before the list premise is written and this commonly identified throughout the write up.

Has the study examined the causality of the problem?

The researchers should consider the following:

1. Provide some background statistics of the existence of the problem within the study sample, for instance companies with institutional investors and their CSR and CSP. This can also validate the problem the study is attempting to solve.

3. Identify whether the research objectives were achieved, if not the limitations.

3. Identify some gaps to be explored by future researchers

7. PLOS authors have the option to publish the peer review history of their article (what does this mean?). If published, this will include your full peer review and any attached files.

Reviewer #1: **Yes: **Halidu Babamu Osman

Reviewer #3: **Yes: **Dr. Jennifer E. Adaletey

---

## [Editor Report · Acceptance letter]

18 Jul 2023

PONE-D-22-35109R1 

The Social Cost of Investor Distraction: Evidence from Institutional Cross-Blockholding 

Dear Dr. Astvansh:

I'm pleased to inform you that your manuscript has been deemed suitable for publication in PLOS ONE. Congratulations! Your manuscript is now with our production department. 

Kind regards, 

on behalf of

Dr. Kofi Mintah Oware 

Academic Editor

PLOS ONE